# Performance Improvement of Ring-Type PZT Ceramics for Ultrasonic Dispersion System

**DOI:** 10.3390/mi11020144

**Published:** 2020-01-28

**Authors:** Young Min Choi, Yang Lae Lee, Eui Su Lim, Mojiz Abbas Trimzi, Seon Ae Hwangbo, Young Bog Ham

**Affiliations:** 1Department of Thermal Systems, Korea Institute of Machinery and Materials, Daejeon 34103, Korea; anaud007@kimm.re.kr (Y.M.C.); yllee@kimm.re.kr (Y.L.L.); eslim@kimm.re.kr (E.S.L.); mat@kimm.re.kr (M.A.T.); 2Department of Plant System & Machinery, University of Science & Technology, Daejeon 34113, Korea; 3Center for Nano-Bio Measurement, Korea Research Institute of Standards and Science, Daejeon 34113, Korea

**Keywords:** lead zirconate titanate, ring-type ceramics, high-intensity focused ultrasound, ultrasonic dispersion system, cavitation effect

## Abstract

This study has been based on the examination of the characterization of ring-type lead zirconate titanate (PZT) ceramics for high-intensity focused ultrasonic dispersion system. The ring-type PZT ceramics were fabricated by the powder molding method. The mechanical properties, dielectric constant, and microstructure of the ceramics were investigated. Consequently, the density of the ceramics was increased with increasing forming pressure while the density of ceramics that were sintered at 1350 °C was decreased due to over-sintering. Furthermore, the mechanical properties were excellent at the higher forming pressure. The dielectric property of the ring-type PZT ceramics was not clearly influenced by the manufacturing and sintering conditions. The abnormal grain growth of the ceramics, however, could be prevented by a lower heating rate in addition to reducing the porosity.

## 1. Introduction

The evolution of electronic materials is closely linked to the development of functional materials, in addition to the growth of the aviation, ship and automobile industries. One of the materials that can be used in these industries is piezoelectric ceramics. Piezoelectric ceramics have drawn considerable interest in recent decades [1]. Today, the materials are important for a variety of applications such as in piezoelectric transducers, sonar, sensors, multilayer capacitors, dielectric resonators, and actuators of the machine due to their outstanding dielectric, ferroelectric, piezoelectric and optoelectronic properties.

Piezoelectric materials are capable of transforming electrical energy into mechanical strain (inverse piezoelectric effect). Ultrasonic transducers operate on the cavitation effect based on the converse effects of piezoelectric materials [2]. The lead zirconate titanate (PZT) is a suitable material for an ultrasonic transducer because of its high piezoelectric constant, relative permittivity and electromechanical coupling coefficient. The high operating frequency above 400 kHz is needed to achieve high-resolution images for high-intensity focused ultrasonic dispersion systems [3,4]. In addition, the substitution of some suitable elements at the A-site (Pb) for different Zr/Ti ratios generally creates distortion, causing modification in the crystal structure, which hence leads to modification in physical properties like microstructure, density, dielectric constant and piezoelectricity. For example, it has been observed in Mn-doped PZT ceramics that there is a decrease in average grain size, and dielectric constant and an increase in dielectric loss with an increase in Mn dopant concentration. The property of the lead-based material can also be enhanced by substituting different rare earth and transition metal ions in the perovskite lattice. The mechanical, piezoelectric and dielectric properties of PZT ceramic compositions near the morphotropic phase boundary (MPB) region with different rare-earth dopants (Gd, Eu, Er, Nd and La) and Bi. The physical properties of these materials can be tailored by doping suitable concentrations of different transition or rare earth metal ions [5,6,7]. On the other hand, to control microstructural parameters, namely, to stabilize the fine-sized state during the transfer from powders to ceramics is to fulfill two key principles. The first one is in the suppression of diffusion mass-transfer and the second principle concerns the formation of ceramics with uniform bi-phase structure. The grain growth during sintering can be limited due to the fact that the grains of each phase pin the boundary of the other phase. This phenomenon helps to avoid the migration of the grain boundary and increase in its size [8,9].

Manufacturing ceramics depends on the characteristics of the raw materials: mean particle size distribution, plasticity, chemical compositions as well as sintering conditions such as final temperature, dwelling time, heating rate and ambient atmosphere. In this paper, the effect of heating rate on the mechanical, morphological and dielectric properties of the ring-type PZT ceramics using powder molding method was studied. Furthermore, the variation of the mechanical and electrical properties of the ceramics was investigated at different forming pressure and sintering temperature selection.

## 2. Materials and Methods

### 2.1. Materials

A commercially available Pb(Zr_0.53_, Ti_0.47_)O_3_ powder (Sunnytec, Suzhou, China) was used in this study. Figure 1 shows the shape morphology of the PZT powder observed by scanning electron microscopy, SEM (Hitachi, Tokyo, Japan). Its shape is overall spherical and agglomerated locally. The PZT powder properties presented (measured by the supplier) longitudinal charge coefficient *d*_33_ = 450 pm/V; planer coupling factor *K*_p_ = 0.71, dielectric constant *ε*_r_ = 2300, Curie temperature *T*_c_ = 260 °C and bulk density = 7.6 g/cm^3^. Table 1 summarizes the chemical composition of commercialized PZT powder by scanning electron microscope with an energy dispersive X-ray spectrometer (Horiba, Kyoto, Japan). The particle size distribution was measured by the laser diffraction particle size distribution analyzer (Beckman Coulter, Brea, CA, USA) as shown in Figure 1.

### 2.2. Sample Preparation and Characterization Methods

The ring-type samples were prepared by the powder molding method using the commercial PZT powder without the binder. The PZT powder was dried at 60 °C for 24 h. The dried PZT powder was dry-pressed at 3.47 MPa, 10.40 MPa and 17.33 MPa into 50 mm ring-shaped disks (inner diameter; 40 mm) in a steel mold. It was pressed by the uniaxial pressure at room temperature. Finally, the green bodies were sintered in air at 1250 °C, 1300 °C and 1350 °C for 2 h; when fixed sintering temperature, the heating rates respectively were 3.0 °C /min, 5.0 °C /min and 10 °C /min to compare the characterization of the ceramics.

The density of sintered ring-type PZT ceramics was measured by the Archimedes method, using distilled water as a liquid media, and the relative density calculated by the division of density and theoretical bulk density (7.6 g/cm^3^). The crystalline structure was examined using X-ray diffraction, XRD (Rigaku, Tokyo, Japan) with Cu Kα source (λ = 0.15405 nm). The samples mounted in reflection mode were analyzed in the ambient atmosphere from 10° to 90° with a step size of 0.02° in 2θ and a scan speed of 10 s/°. The compression strength of the ring-type ceramics was measured by the determination of radial crushing strength (ISO 2739, Geneva, Switzerland) as shown in Figure 2. The maximum load to fracture of each specimen was recorded using a universal testing machine (Testometric, Rochdale, UK) with a cross-head speed of 2.0 mm/min and the compression strength (σ) was calculated using the following equation:σ = F(D − e)/Le^2^,(1)
where F is the maximum load (N); D the outer diameter of the ring (mm); e the thickness of the ring (mm) and L is the length of the ring (mm).

The specimens after the compression testing were cleaned ultrasonically in distilled water and then dried at 60 °C for 1h. Then, the microstructure of fracture surface was examined using a scanning electron microscope (Hitachi, Tokyo, Japan) worked at 15 kV, and the SEM images (at the magnification of 3 k and 10 k) were randomly selected. To measure the hardness of the ceramics surface, we used a Wilson hardness tester Vickers 432 SVD (Buehler, Waukegen, IL, USA) at loads of 5 N.

### 2.3. Polarization and Dielectric Property

The cylindrical surfaces of the ring-type PZT ceramics were uniformly coated with silver paste as electrodes at 600 °C for 10 min. Then for the polarization of the ceramics, each sample was placed in a 120 °C silicone oil bath and a 4.5 kV/mm poling field was applied for 30 min (poling time) to evoke the piezoelectric properties.

The capacitance *C* of the ceramics was measured after cooling the samples sufficiently. The relative dielectric constant *ε*_r_ was calculated from the applied voltage *V*, the electrode area *A* of the sample and the distance *L* of the two substrates, as follows:*C* = *Q*/*V* = *ε*_r_∙*A*/*L*,(2)

## 3. Results and Discussion

### 3.1. Density of the Fabricated Lead Zirconate Titanate (PZT) Ceramics

The optimal sintering temperature of the PZT ceramics was judged by their relative densities sintered at various temperatures and forming pressures. The density of ceramics according to the forming pressure and sintering temperature is shown in Table 2. As the forming pressure increased from 3.47 MPa to 17.33 MPa, the density also increased. Based on our experimental observations, the powder particles gradually get closer to each other under the high pressure and it was accelerated by solid-state sintering of powders at the vitrification stage. However, the density did not increase at 1350 °C of the sintering temperature despite the high forming pressure. With continuous heating, when the temperature reached 1350 °C, the PbO gradually evaporated, leading to the phenomenon of over-sintering due to the relatively low decomposition temperature of PbO compared to the other elements [10,11]. Accordingly, further temperature growth up to 1350 °C leads to the lowest density (i.e., the grains were loosely joined to one another).

Declining the heating rate indicated the shrinkage of bulk PZT ceramic, which can be attributed to the growth of grains by sintering the surrounding powders under the optimal temperature and pressure. Therefore, we had set various heating rates (3 °C/min, 5 °C/min, and 10 °C/min) in this work. As a result, the slower the heating rate, the denser PZT ceramics were obtained. Due to sufficient time of heating, the diffusion from surfaces, lattices, and interfaces resulted in reduced porosity. In order words, it can be clearly seen that with the prolonged holding time, the inner small grains gradually reduced, the larger ones gradually increased, and the gaps between grains were decreased which were pieces of evidence supporting the densification of ceramics.

### 3.2. Structural Analysis

The rhombohedral, tetragonal and perovskite phases—or a mixture of them with variable amounts—are possible due to the fact that commercial PZT powder compositions are near to the morphotropic phase boundary (MPB). Nevertheless, Figure 3 shows the XRD patterns of the PZT ceramics which were manufactured under various sintering conditions and, moreover, the characteristic peaks of PZT revealed that it possessed a typical perovskite structure. The peak splitting is detected in the same peak groups, irrespective of sintering conditions. We used the closed crucible for sintering. Thus, it should be noted that the powders in this work were not calcined and no excess PbO was added to compensate PbO loss from volatilization because all samples were sintered in a crucible. Sintering conditions did not significantly affect the phase formation of the ceramics in this work [12].

### 3.3. Mechanical Property of the Ring-Type PZT Ceramics

In this work, in order to systematically study the effect of forming pressure and sintering conditions on the mechanical property of the ring-type ceramics, PZT powder with MPB composition was selected. Figure 4 shows the compression strength of the ring-type PZT ceramics. It is obvious that the compression strength of the ceramics could be improved by optimal forming pressure and sintering condition. Besides, under the same sintering temperature, the excellent value for the compression strength of the ceramics by a low heating rate was attained. As stated before, it can be explained that increasing the forming pressure in the green body might cause an increase in the compaction density, which led to the increase in the density of the ceramics. Moreover, it can be presented that introducing a low heating rate enhances the sphere close packing and intensifies the network structure of the ceramics. Eventually, the dense ceramics created good mechanical properties. The ring-type PZT ceramics, however, demonstrated brittle fracture when fabricated at the forming pressure lower than about 10 MPa. It was attributed to the formation of internal voids and cracks below critical forming pressure.

Figure 5a shows the variation of the hardness with different forming pressure and final sintering temperature. The samples sintered at 1250 °C were characterized by the lowest hardness ~Hv (Hardness of Vickers) 100, while the highest values of about Hv 280 were demonstrated by samples prepared at 1300°C. The Vickers hardness of the PZT ceramics sintered at 1300 °C reducing by 9% (the compression strength, meanwhile, increased about 10%). Including the values of both the forming pressure and the heating rate at the temperature (1300 °C), the hardness showed a decline with the increase in heating rate as shown in Figure 5b. In general, the diffusion mass transfer processes become accelerated at a low heating rate, leading to the removal of residual pores in the material. Along with pore elimination, selecting the optimal forming pressure and the critical sintering temperature promotes grain growth in ceramics samples. It signifies the increase in hardness with the increase in density. The results confirm that the mechanical properties of PZT ceramics are improved due to the growth of the grain. Similar phenomena are also observed by Fang et al. At the same time, several recent studies [13,14,15,16] demonstrated that the hardness dependent on the grain size is accorded: the increasing grain size leads to a hardening of the material. In other words, it should be noted that, at the low heating rate for sintering, all of them were higher than those at the rapid heating rate, which implies that a more compact structure of the ceramics occurs easily at the low heating rate [17,18,19,20].

### 3.4. Dielectric Property Analysis

The dielectric analysis gives information regarding the dielectric constant from which two electrical characteristics; the insulating nature and the conductive nature of the materials can be understood [21]. The dielectric constant of the ring-type PZT ceramics according to the forming pressure and the heating rate is shown in Table 3. Regarding the fabricated PZT ceramics, the dielectric constant was measured lower than that of the PZT powder (*ε*_r_ = 2300). This could be due to the compounding impurity and the local volatilization of PbO while the samples were sintered [22,23]. Dielectric constant slightly increased with about 2200 at a heating rate of 5 °C/min, but most samples did not exhibit any change in the manufacturing conditions.

### 3.5. Microstructural and Morphology Analysis

Figure 6 shows SEM images for the fracture surface of the PZT ceramics with respect to sintering temperature. The fracture surface shows a combination of inter-granular and trans-granular features. Microstructure analysis has shown that sintering temperature (*Т* = 1250 °C) is not high enough since ceramics sintering is not complete and ceramics are characterized by a large number of open pores as shown in Figure 6a. In Figure 6b, the PZT ceramics sintered at 1300 °C shows some inter-granular pores and predominant inter-granular fracture. The others (Figure 6a,c) present a slightly higher porosity (yellow arrows) than the porosity of the former one. Moreover, the fracture surface of the ceramics sintered at 1300 °C exhibited a high degree of uniformity, while the two kinds of ceramics fabricated at 1250 °C and 1350 °C had rough fracture surface. It can be seen that the grain growth became inhomogeneous when the PZT ceramics were sintered at 1300 °C.

Figure 7 shows the SEM micrographs of the PZT ceramics with different forming pressures and sintering conditions. Using the line intercept method, the mean grain sizes of the ceramics were almost 1.0 ± 0.1 µm. For the PZT ceramics sintered rapidly, it can be seen that the grain growth became inhomogeneous with the mean grain size 1.5 µm (in the range of 0.5–5 µm) and it shows the ceramics manufactured at higher forming pressures more clearly, as shown in Figure 7g. It is reasonable that the fabricated PZT with the optimal forming pressure and the critical sintering temperature has superior state, which is consistent with the density results. The microstructure of the ceramics, moreover, was compact with the decreased heating rate. In the literature reported, the grain is connected with the tendency to combine the same phase grain nearby, because thermally activated processes of mass transfer target at reducing total energy of the system via shrinking grain boundaries [24]. Typically, normal structure (polygon phase) decreases the energy for the nucleation of the PZT crystalline phase as shown in Figure 7f. Therefore, the resulting microstructure is more compact and homogeneous [25].

## 4. Conclusions

The ring-type PZT ceramics were successfully prepared by the powder molding method in this work. The samples were sintered in the range *T* = 1250–1350 °C for 2 h at 3.47–17.33 MPa under the pressure-less furnace. At sintering temperatures of above 1350 °C, the PZT crystals have shown deformation due to over-sintering on the interphase boundaries. It is caused due to a reduction in the density and mechanical properties. Continuously, we investigated the correlations between the mechanical properties and the manufacturing conditions as follows.

The compression strength of ring-type PZT ceramics was confirmed by ISO 2739. The compression strength of the ceramics was enhanced by optimal forming pressure and sintering condition. The ceramics which sintered at a low heating rate had more packed structure and excellent compression strength in particular. The Vickers hardness of the ceramics has been found to be greatly affected by the density. The maximum Vickers hardness of Hv 400 is achieved for ceramics sintered at *T* = 1300 °C (5 °C/min of heating rate). Its result was consistent with the density variations. The observed crystalline phase was identified as the perovskite structure. The peak splitting was detected in the same peak groups, irrespective of sintering conditions due to there being no change in chemical composition. The ceramics sintered at 1300 °C had a homogeneous microstructure. Furthermore, the microstructure of the ceramics was more packed and uniform with a low heating rate. However, we did not find a correlation between manufacturing conditions and dielectric properties, even though the microstructure of the ceramics were changed. While the results are supported by the theory of enhanced mechanical properties, they may not affect the change of the dielectric properties. We assumed that it may be related to the raw materials (PZT powder).

## Figures and Tables

**Figure 1 micromachines-11-00144-f001:**
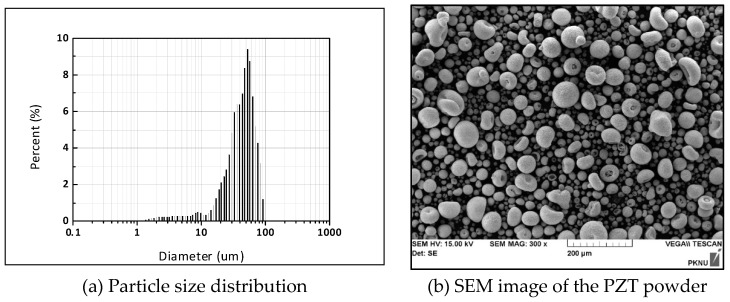
Characteristics of the commercial lead zirconate titanate (PZT) powder: (**a**) Particle size distribution (**b**) scanning electron microscopy (SEM) image of the S-51 PZT powder.

**Figure 2 micromachines-11-00144-f002:**
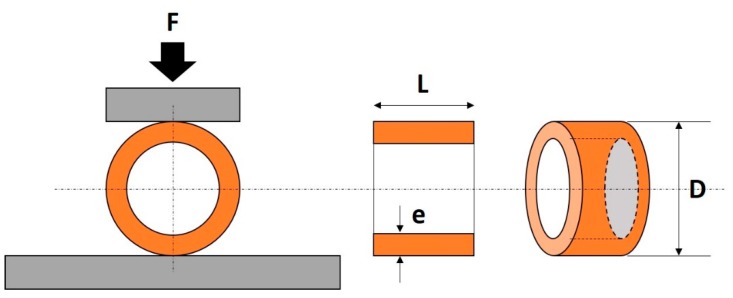
Schematic of compression testing for ring-type ceramics

**Figure 3 micromachines-11-00144-f003:**
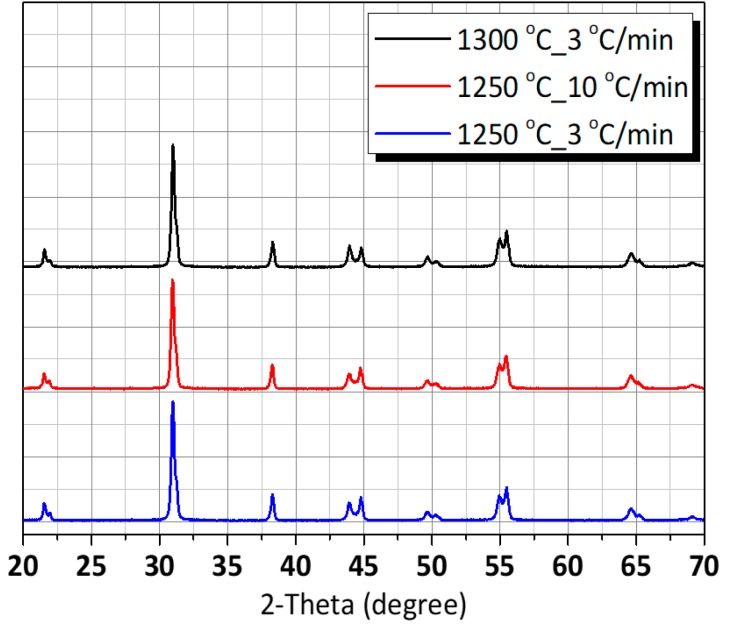
X-ray diffraction (XRD) patterns of the PZT ceramics in accordance with sintering condition.

**Figure 4 micromachines-11-00144-f004:**
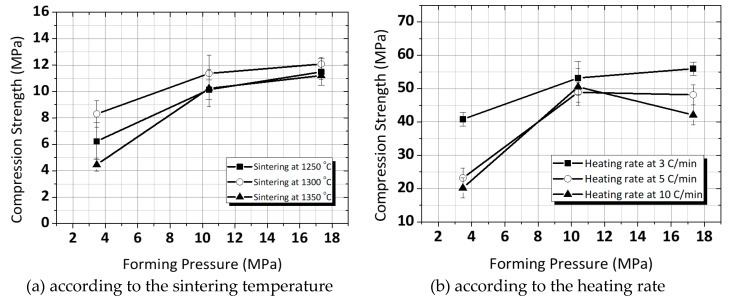
Compression strength of the ring-type PZT ceramics: (**a**) in accordance with sintering temperature (**b**) in accordance with the heating rate (fixed sintering temperature at 1300 °C).

**Figure 5 micromachines-11-00144-f005:**
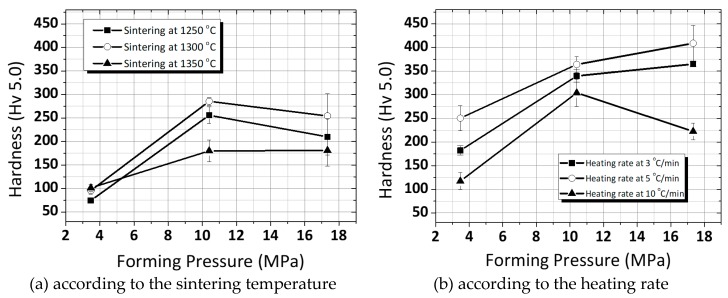
Vickers hardness of the ring-type PZT ceramics (fixed sintering temperature at 1300 °C): (**a**) in accordance with sintering temperature (**b**) in accordance with the heating rate.

**Figure 6 micromachines-11-00144-f006:**
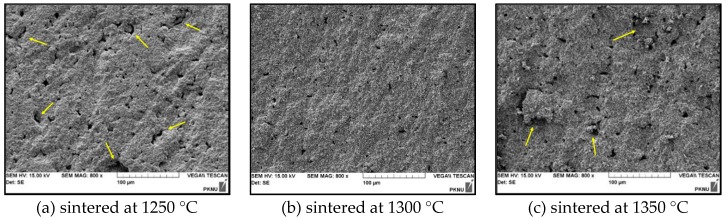
Effect of sintering temperature on the microstructure of the PZT ceramics (fixed 10.40 MPa of forming pressure): (**a**) sintered at 1250 °C (**b**) sintered at 1300 °C and (**c**) sintered at 1350 °C.

**Figure 7 micromachines-11-00144-f007:**
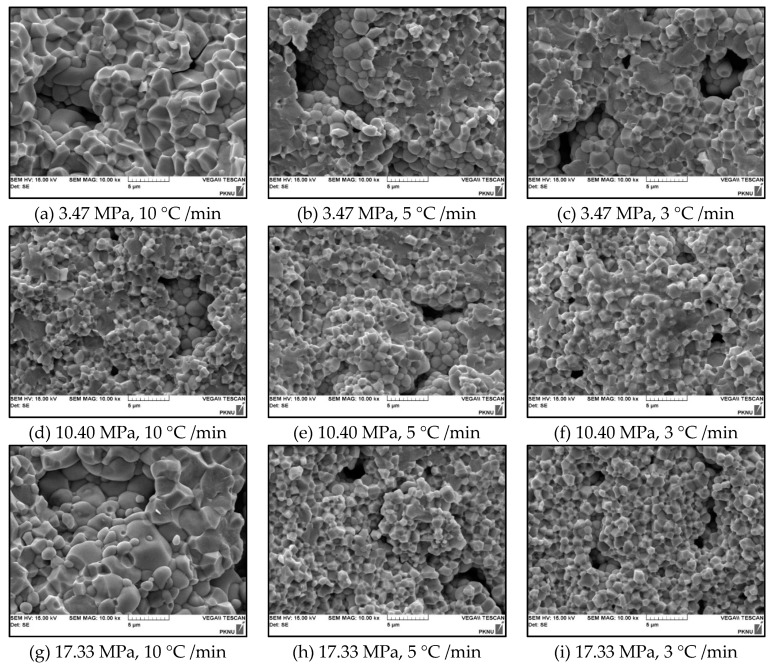
SEM images of fracture surface of the ring-type PZT ceramics according to the forming pressure and the heating rate: (**a**) 3.47 MPa, 10 °C /min (**b**) 3.47 MPa, 5 °C /min (**c**) 3.47 MPa, 3 °C /min, (**d**) 10.40 MPa, 10 °C /min, (**e**) 10.40 MPa, 5 °C/min, (**f**) 10.40 MPa, 3 °C /min, (**g**) 17.33 MPa, 10 °C /min, (**h**) 17.33 MPa, 5 °C /min and (**i**) 17.33 MPa, 3 °C/min.

**Table 1 micromachines-11-00144-t001:** Energy dispersive spectrometer (EDS) results of the lead zirconate titanate (PZT) powder.

Element	Weight (%)	Atomic (%)
O	20.04	69.10
Ti	06.62	07.61
Zr	10.59	06.40
Sn	00.97	00.45
Pb	61.78	16.44
Total	100	

**Table 2 micromachines-11-00144-t002:** Density of the sintered PZT ceramics according to the forming pressure and the sintering condition.

Forming Pressure (MPa)	Sintering Temperature (°C)	Heating Rate (°C/min)	Relative Density (%)
3.47	1250	10	90.33
3.47	1300	10	90.28
3.47	1350	10	88.39
3.47	1300	5	94.07
3.47	1300	3	96.25
10.40	1250	10	95.95
10.40	1300	10	95.40
10.40	1350	10	94.39
10.40	1300	5	97.38
10.40	1300	3	99.78
17.33	1250	10	96.55
17.33	1300	10	96.54
17.33	1350	10	92.08
17.33	1300	5	99.99
17.33	1300	3	99.99

**Table 3 micromachines-11-00144-t003:** Dielectric properties of the ring-type PZT ceramics.

Forming Pressure (MPa)	Heating Rate (°C/min)	Capacitance (nF)	Relative Dielectric Constant (*ε*_r_)
10.40	10	6.18	2124
10.40	5	6.28	2204
10.40	3	5.71	2130
17.33	10	6.46	2078
17.33	5	5.69	2197
17.33	3	5.49	2082

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
