# Peer review of "Performance Improvement of Ring-Type PZT Ceramics for Ultrasonic Dispersion System"

_micromachines, 2020, doi:10.3390/mi11020144_

Round 1

Reviewer 1 Report

In this paper, the authors deals with the manufacturing improvement of PZT rings for ultrasonc dispersion systems.

The paper is well written and easily understandable.

Yet, I found that the presented results, save the mechanical ones, are not innovative enough for PZT based ceramics, where the litterature is huge. This feeling is increased by the fact that the authors conclude themselve that most of the observations are already done in literature. I think that authors could add functional characterizations to enrich their paper.

I have some remarks on the paper:

R1: the authors found a relative density of 99.9% for the ceramics presented in the figure 8 h and i (by the way it could be nice to give a name for each sample) but it seems clear that this ceramic still have residual pores.

R2 : the authors found very similar dielectric constant even if the capacitance change. It can be due to thickness or shape variation but it is confusing. It can be added that the uniformity of the dielectric constant is an unexpected result because we have variation of density (and so of porosity) and of grain size.

Author Response

Dear Reviewer,

I appreciate your valuable comments and queries to strengthen our manuscript.

I am writing to submit our manuscript entitled, “Performance Improvement of Ring-type PZT Ceramics for Ultrasonic Dispersion System” for consideration as a Micromachines Journal research article. We examined the efficacy of using ring-type PZT ceramics as transducer for ultrasonic dispersion device through various experiment and can confirm that the manufactured PZT ring is excellent from commercial one.

I response to your comments, "Please see the attachment"

Again, thank you for giving us the opportunity to strengthen our manuscript with your valuable comments and queries. We have worked hard to incorporate your feedback and hope that these revisions persuade you to accept our submission.

Sincerely,

Young Min Choi

Senior Researcher

Korea Institute of Machinery & Materials

E-mail : anaud007@kimm.re.kr

Reviewer 2 Report

Choi et al. investigate the mechanical properties of the ring shaped PZT as a function of sintering conditions. Generally, the paper is very interesting and the results presentation is very systematic. However, I noticed that several places regarding the data interpretation and mechanism discussion need to be improved. I will reconsider the acceptance after my following concerns are ALL addressed.

There are lots of typos throughout the text. The authors argue that the relatively low density of the 1350C sintered sample is due to the PbO loss. Is there any direct evidence? Figure 4 does not show the XRD of the 1350C sample. As it could help, please incorporate it at least in the response letter. The authors explain the grain size dependence of the hardness using inverse Hall Petch effect. As Hall Petch effect is primarily a dislocation-related theory, the author should justify how it is applicable to ceramic. If justified, the authors also need to confirm that different sintering condition does not affect the total quantity of the dislocations.  Do the authors think that some secondary phases are observable in Figure 7c? Have they checked the EDS? Although the authors have the SEM image, they don't provide any quantitative analysis on the grain size/grain size distribution. For instance, Figure 8g looks very distinctive. To my naked eyes, the average grain size is decreasing as the forming pressure rises (b to e to h and c to f to i). However, from a to d to g, the trend is decrease and then increase. 

Author Response

(The authors gave the same response as above.)

Round 2

Reviewer 2 Report

All my concerns have been well addressed, except for the Hall-Petch effect. First, I think Hall-Petch effect is applicable while plastic deformation takes place (dislocations migrate). The author should discuss how it is applicable to ceramics which has no ductility. The two new references are on metal. Second, Hall-Petch effect manifests in the linear relation between the hardness and the grain size. But in the current version, there is no such data. So, I really don't think the author can articulate that their ceramic exhibits Hall-Petch behavior. 

Author Response

Dear Reviewer

Thank you for your valuable comment. First of all, it is true that this manuscript is an interesting paper but lack of scientific novelties. This paper, however, does generate a detail study of the ring-type piezoelectric ceramics for ultrasonic dispersion systems, which would be useful for researchers working in the same topic.

We had suggested that the hardness dependent on the grain size is accorded with the reverse Hall–Petch effect. However, we agree with you and have incorporated the suggestion throughout our paper. We removed the Hall-Petch equation and hope that the deletion clarifies the points we attempted to make. And we have also added new references [14-17], which outlines the compression-related microhardness of ceramics: implying the linear dependence between the microhardness and the inverse square root of grain size. We hope that our edits and the responses we provide satisfactorily address all the issues and concerns you have noted.

Special thanks again to you for your valuable comments and precious time!

Best Regards,

Young Min Choi

Senior Researcher

Korea Institute of Machinery and Materials
